

# Size-age population structure of an endangered and anthropogenically introgressed northern Adriatic population of marble trout (*Salmo marmoratus* Cuv.): insights for its conservation and sustainable exploitation

Gianluca Polgar[1], Mattia Iaia[1], Paolo Sala[1], Tsung Fei Khang[2,3], Silvia Galafassi[1], Silvia Zaupa[1] and Pietro Volta[1]

[1] Water Research Institute (IRSA)—CNR, Verbania Pallanza, VB, Italy
[2] Institute of Mathematical Sciences, Faculty of Science, Universiti Malaya, Kuala Lumpur, Malaysia
[3] Universiti Malaya Centre for Data Analytics, Universiti Malaya, Kuala Lumpur, Malaysia

Corresponding author
Gianluca Polgar,
gianluca.polgar@gmail.com

## ABSTRACT

Salmonid species are main actors in the Italian socio-ecological landscape of inland fisheries. We present novel data on the size-age structure of one of the remnant Italian populations of the critically endangered marble trout *Salmo marmoratus*, which co-occurs with other stocked non-native salmonids in a large glacial river of the Lake Maggiore basin (Northern Italy-Southern Switzerland). Like other Italian native trout populations, the Toce River marble trout population is affected by anthropogenic introgression with the non-native brown trout *S. trutta*. Our sample includes 579 individuals, mainly collected in the Toce River main channel. We estimated the length-weight relationship, described the population size-age structure, estimated the age-specific growth trajectories, and fit an exponential mortality model. A subset of the sample was also used to measure numerical and biomass density. The estimated asymptotic maximum length is ~105 cm total length (*TL*). Mean length at first maturity is ~55 cm *TL*, and mean length at maximum yield per recruit is ~68 cm *TL*. Approximately 45–70% of the population are estimated to die annually, along with a fishing annual mortality of ~37%, with an exploitation ratio of ~0.5. The frequency distribution of length classes in a sample collected by angling shows that ~80% of the individuals that could be retained according to the current recreational fishing regulations likely never reproduced, and large fish disproportionally contributing to recruitment are fished and retained. We identify possible overfishing risks posed by present regulations, and propose updated harvest-slot length limits to mitigate such risks. More detailed and long-term datasets on this system are needed to more specifically inform the fishery management and monitor the effects of any change in the management strategy on the size-age structure of the marble trout population of the Toce River.

## INTRODUCTION

In the inland waters of industrialized countries, sport fishing is an extremely popular leisure activity that attracts social, cultural, and economic interests; salmonids make up the backbone of these fisheries (*Unfer & Pinter, 2018*; *Brown et al., 2019*; *Arlinghaus et al., 2019*). Unsustainable management of salmonid stocks is a widespread concern. Overharvesting and poorly designed hatchery and stocking programs fuel the spreading of invasive and interfertile non-native taxa that compete and hybridize with native populations (*e.g.*, *Naish et al., 2007*). These anthropogenic impacts interact synergistically with habitat destruction, habitat degradation, and climate change, thus affecting the population size-age structure of native salmonids through density-dependent regulating mechanisms (*e.g.*, *Marschall & Crowder, 1996*; *Vincenzi et al., 2007a*; *Ayllón et al., 2019*).

The marble trout *Salmo marmoratus* Cuvier, 1829 is a large salmonid exceeding one meter in total length (*TL*). It is characterized by a marbled coloration and is phylogenetically distinct from other *Salmo* species (*e.g.*, *Pustovrh, Sušnik Bajec & Snoj, 2011*; *Pustovrh, Snoj & Sušnik Bajec, 2014*; *Segherloo et al., 2021*). Among salmonids, marbled coloration traits are also present in (i) hybrids between *S. marmoratus* and domesticated stocks of North-European brown trout (*Delling et al., 2000*; *Djurdjevič et al., 2019*; *S. trutta* Linnaeus, 1758 *sensu Segherloo et al., 2021*), (ii) at least one population of *S. trutta* (Otra River, Norway; *Delling, 2002*), and (iii) hybrids between salmonid species of different genera (*Miyazawa, Okamoto & Kondo, 2010*). *S. marmoratus* includes two geographically distinct populations, in the Northern and Southern Adriatic regions (*Delling et al., 2000*; *Polgar et al., 2022* and references therein). In the North Adriatic region, this species is distributed in the middle and lower tracts of the orographic left tributaries of the Po River, a few of the small orographic right tributaries of the Po River in the Southwestern Alps, and several large rivers flowing into the Adriatic Sea (Fig. 1; *Sommani, 1960*; *Meraner & Gandolfi, 2018*; *Lobón-Cerviá et al., 2019*; *Splendiani et al., 2020*; *Merati, Pascale & Perosino, 2021*). In the upper drainage of the Slovenian Soča River, in rhithral conditions, several small, genetically pure (*Berrebi et al., 2000*; *Snoj et al., 2000*; *Sušnik Bajec et al., 2015*) and extremely differentiated (*Fumagalli et al., 2002*) populations live in small karstic streams isolated by insurmountable waterfalls. These populations are periodically exposed to drastic demographic and genetic bottlenecks induced by stochastic and catastrophic flood events (*e.g.*, *Pujolar et al., 2016*). Although earlier studies (*e.g.*, *Boniforti, 1870*; *Pavesi, 1873*; *Delpino, 1935*), including its original description (*Cuvier, 1829*), recorded this species also in subalpine lakes, there is presently no scientifically sound documented record of lacustrine or lacustrine-adfluvial populations (*Meraner & Gandolfi, 2018*). It is not possible to know whether these records are native marble trout populations, straying stocked individuals, lacustrine morphs of non-native *Salmo* species, hybrids between native and non-native taxa with divergent behavioral patterns, or native and extinct *Salmo* species (*e.g.*, *D'Ancona & Merlo, 1959*; *Behnke, 1972*; *Giuffra, Guyomard & Forneris, 1996*). Confounding factors include the pervasive and frequent translocations

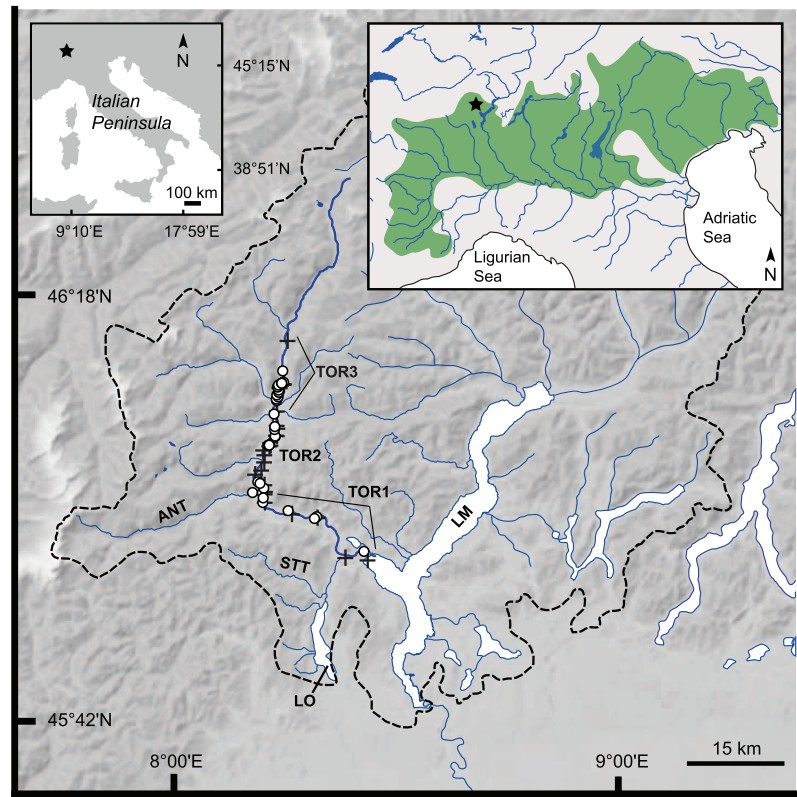

**Figure 1 Study area: Toce River lower, middle, and upper basin.** Dashed black line: Lake Maggiore's drainage divide (*Barbanti, 1994*); upper left inset: position of the study area (star symbol) in the Italian Peninsula; upper right inset: geographical distribution of Italian Northern Adriatic populations of *S. marmoratus* (green area), and position of the study area (star symbol). *ANT*, Anza Torrent; *LM*, Lake Maggiore; *LO*, Lake Orta; *STT*, Strona Torrent; *TOR-1* to *TOR-3*, lower, middle, and upper Toce River, respectively; black crosses: electrofishing sample (A); white dots: angling sample (B). Google Earth Pro©2021 v.7.3, Garmin BaseCamp v.4.7, and QGIS v.3.16 were used to position GPS waypoints on the map.

that have occurred in this territory in historical times (*e.g.*, *Delpino, 1935*), the lack of genetic data, and the strong convergent traits of lacustrine and lacustrine-adfluvial *Salmo* species (*e.g.*, *Snoj et al., 2010*).

Scientific investigations of marble trout populations in the North Adriatic region have been historically affected by the presence of anthropogenic introgressive hybridization between the marble trout and the non-native and interfertile Atlantic brown trout *S. trutta*, introduced by stocking activities (*Splendiani et al., 2016*, *2019*). Phenotypic hybrids between *S. trutta* and *S. marmoratus* were recorded since the early 20[th] century (*e.g.*, *Delpino, 1935*; *Pomini, 1940*; *Sommani, 1948*). Subsequent molecular studies confirmed the presence of hybridization between these two species in the wild (*Giuffra, Guyomard & Forneris, 1996*; *Lucarda et al., 2000*; *Zerunian, 2003*; *Jug, Berrebi & Snoj, 2005*; *Turin, Zanetti & Bilò, 2006*; *Lucarda, 2007*; *Meldgaard et al., 2007*; *Meraner et al., 2010*; *Meraner & Gandolfi, 2018*). It is still unclear whether these hybrids are adaptively advantaged or disadvantaged (positive or negative heterosis, respectively) either in sympatry or allopatry with 'genetically pure' marble trout, though possible ecological differences and interactions

between genetically discriminated hybrids and 'pure' individuals have been explored (*Meldgaard et al., 2007*; *Meraner et al., 2010*; *Simčič, Jesenšek & Brancelj, 2017*). Several ecological studies, including some early investigations on population density, size-age structure, and fishing pressure, phenotypically discriminated between pure and hybrid individuals within populations, analyzing them separately (*Alessio et al., 1990*; *Jelli & Duchi, 1990*; *Jelli, Alessio & Duchi, 1996*; *Turin, 2000*). However, such analyses are scarcely informative, due to the substantial overlap between the coloration pattern of *S. trutta* × *S. marmoratus* hybrids and that of both parental species, which prevents from phenotypically discriminating hybrids from 'genetically pure' marble trout individuals, as evidenced by morphological, genetic, and transcriptomic studies (*Delling et al., 2000*; *Meraner et al., 2010*; *Chiesa et al., 2016*; *Djurdjevič et al., 2019*; *Gandolfi et al., 2020*). Studies that phenotypically determined *S. marmoratus* without discriminating between pure and hybrid phenotypes offer a synthetic scenario of introgressed marble trout populations, though not discerning between possibly different ecological contributions of hybrid and non-hybrid individuals. However, none of these studies examined the size-age structure of a population (*e.g.*, *Lorenzoni et al., 2012*; *Marchi et al., 2017*). When genetically pure wild populations were found (Soča River headwaters), some ecological studies examined them *in situ*, investigating density and age structure (*Šumer, Leiner & Povž, 2001*) or their trophic ecology, including interactions with non-native trout species (*Vincenzi et al., 2011*; *Musseau et al., 2015*, *2017*, *2019*). Other studies used hatchery-bred progeny from broodstock obtained from these populations to conduct field experiments and investigate plastic or adaptive density-dependent responses to catastrophic events and climate change (*e.g.*, *Crivelli et al., 2000*; *Vincenzi et al., 2007a*, *2007b*, *2010*, *2012a*, *2012b*, *2014*; *Vincenzi, Jesensek & Crivelli, 2020*), or to measure metabolic potential, respiratory rate, and ecological tolerance in the laboratory (*Simčič, Jesenšec & Brancelj, 2015a*, *2015b*, *2017*).

Within the North Adriatic region, in northern Italy, the marble trout is a subendemic species with substantial cultural, conservation and economic importance (*Meraner & Gandolfi, 2018*). This species was first included in the Annex II of the EU Habitats Directive (*European Union, 1992*), and then in an action plan for freshwater fish conservation, which classified the Italian populations as 'IUCN Endangered' (*Zerunian, 2003*). However, in spite of past efforts to facilitate a reduction of the angling pressure through fishing regulations and supporting breeding programs, which resulted in moderate numerical increases of the populations since the 1980s (*Turin, Zanetti & Bilò, 2006*; *Lucarda, 2007*), Italian populations have been subsequently classified as 'Critically Endangered' by the Italian IUCN Red List, predicting an 80% decline, due to introgressive hybridization with the Atlantic *S. trutta* and habitat degradation (*Bianco et al., 2013*). The massive introduction of *S. trutta* dramatically impacted marble trout populations in northern Italy, and no 'genetically pure' populations are known. However, panmixia has been observed in very few cases and genetically pure individuals are present in most populations (*Meraner & Gandolfi, 2018*), possibly due to a partial overlap of the spawning period of the two parental species and to local differences in the relative size of introduced non-native stocks and native populations (*Meraner et al., 2010*; *Meraner & Gandolfi,*

*2018*). Like many other salmonid species, Italian marble trout populations also suffer habitat destruction and degradation from damming, channelization, water withdrawal, hydropeaking events, extraction of riverbed materials, artificial embankments, degradation or destruction of the riparian vegetation, and climate change (*Zerunian, 2003*; *Becciu & Dresti, 2015*; *Dresti et al., 2016*; *Saidi et al., 2014*, *2018*), with possible but yet uninvestigated synergistic effects on population structure.

In the VCO Province of Piedmont, within the subalpine catchment of Lake Maggiore (Fig. 1), the Toce River hosts an ecologically and economically important trout fishery, where *Salmo marmoratus* co-occurs with other stocked non-native salmonids, including non-native *S. trutta*. Recent investigations confirmed the presence of hybridization between *S. marmoratus* and *S. trutta* in the Toce River (*Gibertoni et al., 2014*). No other hybridization events between native and non-native species have yet been documented in this system. Furthermore, no stocks of *S. trutta* with marbled coloration traits and no hybrids between salmonid species of different genera with marbled coloration were found in this system. Therefore, it is reasonable to assume that trout with marbled coloration patterns here only include 'genetically pure' *S. marmoratus* and *S. trutta* × *S. marmoratus* hybrids or variably introgressed individuals. On the other hand, the absence of marbled coloration traits in introgressed individuals cannot be ruled out.

Here, we analyze trout specimens with marbled coloration elements of the anthropogenically introgressed marble trout population of the Toce River to provide novel baseline data on its size-age structure. To this aim, we: (1) estimate biomass density and numerical density; (2) estimate length-mass relationships, size-age structure and body growth trajectories, length at first maturity, length at maximum yield per recruit, mortality; and (3) illustrate the possible effects of present minimum-length harvest limits *(MLL)* regulations on the population's length structure, relative to an alternative fishing strategy. We then discuss future research developments to address possible management options, with each option appealing to different conservation, societal and political priorities.

## MATERIALS AND METHODS

### Study area, fish sampling, taxonomic discrimination, and aging

The Toce River (length 83.6 km; catchment area ~1,780 km$^2$, average slope 2.4%; *Regione Piemonte, 2021*; Fig. 1; Note S1) is one of the main tributaries of Lake Maggiore, rising from glacier valleys at ~1,720 m above sea level (a.s.l.; *Geoportale Piemonte, 2021*). It is located in the Italian North-western Alps (*Marazzi, 2005*), in the Padano-Venetian ichthyogeographic region (*Bianco, 1998*). In the VCO Province of Piedmont, a yearly average of ~3,600 resident and non-resident anglers buy fishing licenses (main national angling association: G. M. Bertoia, 2021, personal communication). Trout stocking is a profitable activity, with investments, resources, and satellite activities revolving around recreational inland fishing, including building and maintenance of the hatcheries, the trading of hatchery stocks, fishing licenses and membership issued by angling associations, angling competitions, and tourism (P. Volta, 2022, personal observation).

Three types of stocking management (*Lorenzen, Beveridge & Mangel, 2012*) are practiced in VCO: culture-based fisheries of non-native species (production-oriented),

including (i) a put-grow-and-take fishery of *S. trutta* and of *Salmo ghigii* Pomini, 1941 (*sensu Polgar et al., 2022*), variably introgressed with *S. trutta* (*Splendiani et al., 2019*) and commercially referred to as 'Mediterranean brown trout' (mainly Lake Maggiore, upper Toce River and most tributaries); (ii) a put-and-take fishery of *Oncorhynchus mykiss* Walbaum, 1792 (upper Toce River and some of its tributaries); and (iii) a stock-enhancement fishery (supportive breeding) of *S. marmoratus* (mainly Lake Maggiore and Lake Orta, middle and lower tracts of the Toce River, and several tributaries).

Fishing and fishery management is delegated by the VCO Province to the local branch of the F.I.P.S.A.S., and is implemented through input (*i.e.*, closed areas, closed fishing seasons, gear restrictions, licensing), and output (*i.e.*, length-based harvest limits, daily and annual bag limits) control regulations (*Verbano Cusio Ossola Province, 2021*). Stocking is regulated by the Province and (i) performed by the F.I.P.S.A.S. in public waters; (ii) outsourced to private contractors and local fishing associations in private waters; and (iii) performed by hydroelectric companies, contractually obligated to stock trouts for environmental compensation, which buy live fish from private local and non-local hatcheries (P. Volta, 2022, personal observation). Fish stocking in protected areas (Natura 2000 network; *Ministry of Ecological Transition of Italy (MTE), 2020*) is further regulated by national laws (*Ministry for the Environment, Land and Sea of Italy (MATTM), 2020*), compelling the managers of protected areas to follow strict assessment procedures called VINCA ('Valutazione di Incidenza') for the assessment of the potential impact of stocked fish on the native biodiversity and habitats of high naturalistic value. Monitoring and enforcement of regulations is rarely and scarcely implemented (P. Volta, 2022, personal observation). Genetic analyses of domesticated marble-trout stocks are seldom conducted, and identification of trout breeders is routinely based on phenotypes.

Since 2019, the marble trout *MLL* in the Toce River is 40 cm *TL* (before 2019: 35 cm *TL*). The daily bag limit, including hybrids, is of two individuals (10 individuals per year) (*Verbano Cusio Ossola Province, 2021*). No experimental data are publicly available on the population dynamics, structure, or fishing pressure of this fishery. Consistently, the ecology of the marble-trout population of the Toce River is virtually unknown. The only available study provides limited information on a possibly lacustrine-adfluvial (pelagic) morph of *S. marmoratus* (*Gibertoni et al., 2014*).

Fish sampling was conducted from May 2016 to November 2020. Three areas of the Toce River were sampled: (i) the lower tract, plus one site at the confluence with the Strona Torrent (*TOR1*); (ii) the middle tract, plus one site at the confluence with the Anza Torrent (*TOR2*); and (iii) the upper tract (*TOR3*); one site was also sampled in Lake Maggiore, less than 1 km off the Toce River mouth, and was included in the *TOR1* dataset (Tables 1, S1, S2; Fig. 1).

Two sampling methods were used: electrofishing (A) and angling (B). In the typical environmental conditions of the study area, both methods can be conducted at similar minimum depths (~0.2–0.5 m). In contrast, the maximum depth is ~1.5 m for electrofishing, and ~4 m for rod-and-line angling (nymph-fly, dry-fly, spinning). Therefore, the use of both methods enabled the surveying of a larger portion of the marble

**Table 1** Sample of *S. marmoratus* from the Toce River Basin.

| Year | Ar | S | n | TL (cm) Range | TL (cm) Mean ± s.d. | W (g) Range | W (g) Mean ± s.d. | Age (yr) Range | Age (yr) Mean ± s.d. | TL-W | DENS | TL-Age |
|---|---|---|---|---|---|---|---|---|---|---|---|---|
| 2016 | TOR3 | B | 2 | 33.0, 41.0 | 37.0 | 300, 626 | 463 | 3.2, 4.2 | 3.7 | 2 | 0 | 2 |
| 2017 | TOR1 | A | 27 | 7.3–65.0 | 21.4 ± 10.8 | 3–2,749 | 195 ± 519 | 1.1–2.1 | 1.4 ± 0.5 | 4 | 27 | 4 |
| 2017 | TOR2 | A | 139 | 6.3–58.0 | 20.6 ± 9.9 | 2–1,941 | 147 ± 250 | 0.3–6.0 | 3.0 ± 1.3 | 62 | 135 | 37 |
| 2017 | TOR3 | A | 20 | 14.0–38.5 | 24.9 ± 6.1 | 25–380 | 167 ± 101 | 2.3–5.3 | 3.0 ± 1.0 | 16 | 20 | 9 |
| 2017 | TOR3 | B | 2 | 29.5, 30.5 | 30.0 ± 0.7 | 214, 230 | 222 | 3.3, 3.4 | 3.4 | 2 | 0 | 2 |
| 2017 | TOR3 | A, B | 22 | 14.0–38.5 | 25.3 ± 6.0 | 25–380 | 172 ± 97 | 2.3–5.3 | 3.1 ± 0.9 | 18 | 20 | 11 |
| tot2017 | | A, B | 188 | 6.3–65.0 | 21.3 ± 9.7 | 2–2749 | 157 ± 291 | 0.3–6.0 | 2.9 ± 1.3 | 84 | 182 | 52 |
| 2018 | TOR2 | B | 9 | 20.7–37.0 | 27.9 ± 6.0 | 101–447 | 216 ± 122 | 1.4–3.4 | 2.4 ± 0.7 | 9 | 0 | 9 |
| 2018 | TOR3 | B | 8 | 23.1––31.4 | 27.5 ± 3.2 | 108–329 | 204 ± 80 | 2.4–3.5 | 2.9 ± 0.5 | 8 | 0 | 8 |
| tot2018 | | B | 17 | 20.7–37.0 | 27.8 ± 4.8 | 101–447 | 210 ± 101 | 1.4–3.5 | 2.7 ± 0.7 | 17 | 0 | 17 |
| 2019 | TOR3 | A | 20 | 11.0–29.5 | 16.8 ± 5.0 | 9–227 | 56 ± 56 | 1.3–3.3 | 1.7 ± 0.7 | 20 | 20 | 13 |
| 2019 | TOR3 | B | 1 | 26.8 | – | 191 | – | 3.3 | – | 1 | 0 | 1 |
| tot2019 | | A, B | 21 | 11.0–29.5 | 17.3 ± 5.4 | 9–227 | 62 ± 62 | 1.3–3.3 | 1.8 ± 0.8 | 21 | 20 | 14 |
| 2020 | TOR1 | A | 51 | 10.6–43.1 | 17.3 ± 6.6 | 10–737 | 72 ± 123 | na | na | 48 | 51 | 0 |
| 2020 | TOR2 | A | 46 | 14.2–80.0 | 31.7 ± 16.0 | 26–4,800 | 573 ± 862 | 1.3–10.7 | 3.5 ± 2.1 | 32 | 44 | 33 |
| 2020 | TOR3 | A | 163 | 7.0–80.0 | 17.8 ± 9.6 | 3–4,689 | 120 ± 434 | 0.5–10.7 | 1.9 ± 1.4 | 156 | 161 | 90 |
| 2020 | TOR1 | B | 10 | 19.8–85.0 | 34.7 ± 19.1 | 77–4,600 | 720 ± 1,381 | 1.2–9.8 | 2.7 ± 2.5 | 10 | 0 | 10 |
| 2020 | TOR2 | B | 61 | 19.0–58.0 | 33.1 ± 9.6 | 73–2,100 | 505 ± 490 | 1.3–6.4 | 2.8 ± 1.2 | 61 | 0 | 60 |
| 2020 | TOR3 | B | 20 | 19.5–31.0 | 24.1 ± 2.8 | 75–310 | 155 ± 65 | 1.4–3.3 | 2.1 ± 0.6 | 20 | 0 | 17 |
| 2020 | TOR1 | A, B | 61 | 10.6–85.0 | 20.2 ± 11.5 | 10–4,600 | 179 ± 597 | 1.2–9.8 | 2.7 ± 2.5 | 58 | 51 | 10 |
| 2020 | TOR2 | A, B | 107 | 14.2–80.0 | 32.5 ± 12.7 | 26–4,800 | 534 ± 673 | 1.3–10.7 | 3.0 ± 1.6 | 93 | 44 | 93 |
| 2020 | TOR3 | A, B | 183 | 7.0–80.0 | 18.5 ± 9.3 | 3–4,689 | 124 ± 410 | 0.5–10.7 | 2.0 ± 1.3 | 176 | 161 | 107 |
| tot2020 | | A, B | 351 | 7.0–85.0 | 23.0 ± 12.5 | 3–4,800 | 258 ± 565 | 0.5–10.7 | 2.5 ± 1.6 | 327 | 256 | 210 |
| totA | | | 466 | 6.3–80.0 | 20.4 ± 10.8 | 2–4,800 | 171 ± 438 | 0.3–10.7 | 2.4 ± 1.6 | 338 | 458 | 186 |
| totB | | | 113 | 19.0–85.0 | 30.8 ± 9.9 | 73–4,600 | 409 ± 563 | 1.2–9.8 | 2.7 ± 1.3 | 113 | 0 | 109 |
| totTOR1 | | | 88 | 7.3–85.0 | 20.5 ± 11.3 | 3–4,600 | 184 ± 572 | 1.1–9.8 | 2.3 ± 2.2 | 62 | 78 | 14 |
| totTOR2 | | | 255 | 6.3–80.0 | 25.9 ± 12.5 | 2–4,800 | 312 ± 509 | 0.3–10.7 | 3.0 ± 1.5 | 164 | 179 | 139 |
| totTOR3 | | | 236 | 7.0–80.0 | 19.5 ± 9.1 | 3–4,689 | 128 ± 365 | 0.5–10.7 | 2.1 ± 1.2 | 225 | 201 | 142 |
| Total | | | 579 | 6.3–85.0 | 22.5 ± 11.4 | 2–4,800 | 218 ± 474 | 0.3–10.7 | 2.5 ± 1.5 | 451 | 458 | 295 |

**Note:**

*Year*, year of survey; *Ar*, sampled water bodies, including the lower tract of the Toce River plus one site at the confluence with the Strona Torrent, and one site in Lake Maggiore (*TOR1*), the middle tract of the Toce River, plus one site at the confluence with the Anza Torrent (*TOR2*), and the upper tract of the Toce River (*TOR3*) (Fig. 1); *S*, type of sample (A: electrofishing, B: angling); *n*, number of sampled individuals; *TL (cm)*, total length; *W (g)*, wet-mass; *Age*, age in years; *na*, not available data (for all measurements, s.d. was computed when $n \geq 4$); *TL-W*, number of individuals used to estimate the length-weight regression; *DENS*, number of individuals used to estimate numerical density and biomass density; *TL-Age*, number of individuals used to describe the population age structure, estimate age-specific growth trajectories, and fit an exponential mortality model. *Total*, total number of sampled individuals, overall ranges and means ± s.d.; *Subtotal A, B*, individuals in samples A, B, ranges and overall means ± s.d. The three subsamples used to conduct the analyses (*TL-W*, *DENS*, *TL-Age*) were mainly collected in 2017 and 2020. Raw data set in Table S1.

trout habitat. Additionally, this also allowed us to experimentally observe the potential impact of recreational angling on the population, based on current fishing regulations. In one field survey, lake trolling near the river mouth was also performed at a depth of 10–40 m. Electrofishing surveys were arranged in georeferenced transects, while angling surveys were defined as areas (0.6 ± 1.0 standard deviation: s.d. km$^2$) within which temporally subsequent angling sessions were conducted.

Sample A was collected during daytime, proceeding on foot along transects made in shallow water along riverbanks, with a built-in-frame EL64GII electrofishing device (Scubla aquaculture, 3.5 KW, 600 V, DC current). This device has a copper cathode (width 2 cm, length 300 cm), a steel ring anode (thickness 0.8 cm, diameter 50 cm), and an effective electrical field of ~1 m$^3$ (*Copp & Garner, 1995*). Linear transects were georeferenced with a GPS device for 31 out of 36 surveys. Garmin BaseCamp v.4.7 and QGIS v.3.16 were used to analyze GPS data and calculate transect lengths. The device was swept by the operator on both sides of the path, and operated at depths of ~50–150 cm (median value = 100 cm). Therefore, the sampling effort (sampled water volume during each surveyed meter) was calculated as $L \times 2$ m$^3$, where $L$ is the transect length (Tables S1, S2).

For all sampling methods, the caught fish were anaesthetized in a water volume of 10 l by adding 2 ml of a 1:5 emulsion of clove oil (eugenol) in 96% ethanol. When the fish lost its righting reflex, it was rapidly positioned on an ichthyometer, photographed in lateral view, and measured (*TL*: total length, in cm, to the nearest mm; *W*: wet body mass, to the nearest g). A small sample of dorsolateral scales was also collected. Manipulation was reduced to a minimum, and the fish was immediately released after measurements. Sampled individuals were *Salmo* specimens with marbled elements (spots with amoeboid shape) in their coloration pattern, thus including 'genetically pure' *S. marmoratus*, *S. trutta* × *S. marmoratus* hybrids, and variably introgressed individuals (*Delling, 2002*; *Djurdjevič et al., 2019*; *Lorenzoni et al., 2019a, 2019b*; *Polgar et al., 2022*; Fig. S1).

No information was found in the literature about scale reading for age determination in *S. marmoratus*, therefore we used published information on sea trout (*Elliot & Chambers, 1996*; Note S1). Age determination was based on the knowledge of the (i) breeding season of *S. marmoratus* in the study area (M. Iaia, 2021, personal observation); (ii) number of degree-days needed for hatching (*Loro & Zanetti, 1991*; *Turin, 2000*; *Zerunian, 2004*; *Kottelat & Freyhof, 2007*; M. Iaia, 2021, personal observation); (iii) number of degree-days needed for the first development of dorsolateral scales, after the fry leave the redd and start feeding (*Ericksen, 1999*; *Kottelat & Freyhof, 2007*); and (iv) measured monthly surface-water temperatures of the Toce River (Note S1; Fig. S2).

Pit-tags (*Pacific States Marine Fisheries Commission, 2014*) were used to individually mark captured and released individuals in 2019 ($n = 19$, *i.e.*, 73% of the sample) and 2020 ($n = 236$, *i.e.*, 67% of the sample). The overall relative frequency of recaptured individuals is 0.023; four individuals were recaptured twice, two individuals in 2019, one individual in 2020, and one individual in 2019 and 2020; and two individuals were recaptured thrice, both twice in 2019 and once in 2020. These six individuals were included only once in all the analyses. Between 2016 and 2018 (35% of the total sample; Table 1) no individual was pit-tagged; however, due to the small frequency of recaptures, we assumed that the impact of possible recaptures during this period on the analyses was not statistically significant.

Overall, electrofishing and angling comparably contributed to the whole sample, including 56% and 44% of the total number of surveys ($n = 64$), respectively; most surveys (89%) were conducted in 2017, 2019, and 2020, when both sampling methods were adopted in the same year (Table S2). No electrofishing surveys (sample A) could be

conducted in 2016 and 2018, and in both *TOR1* and *TOR2* in 2019; no angling surveys (sample B) could be conducted in *TOR1* and *TOR2*, both in 2017 and 2019 (Table S2). Sample A was used to measure numerical density and biomass density; both samples A and B were used to estimate the length-weight relationship, describe the population age structure, estimate age-specific growth trajectories, and fit an instantaneous mortality model ("Fish density, length-mass relationship, length-age analysis, mean length at first maturity, optimum length, and mortality"; Tables 1, S1).

## Fish density, length-mass relationship, length-age analysis, mean length at first maturity, optimum length, and mortality

A length-weight regression for the Toce River marble trout population was estimated using subsets of sample A ($n = 338$) and sample B ($n = 113$) with measured *TL* and *W* data (Tables 1, S1), using the following allometric model:

$$\log_{10}(W) = b \cdot \log_{10}(TL) + \log_{10}(a) \tag{1}$$

A subset of sample A was used to measure numerical density (ind. m$^{-3}$) and biomass density (g m$^{-3}$) per river tract, when georeferenced linear transects were available (31 surveys, $n = 458$; Table 1). Within this subset, since wet-mass data are not available for 124 individuals, Eq. (1) was used to impute missing data (Table S1).

Other subsets of samples A ($n = 186$) and B ($n = 109$) with available length measurements and determined age (Table S1) were used for length-age analyses. Age classes were defined using time intervals of 1 year, excluding the lower limit value of each interval (*i.e.*, 0 < class 0 ≤ 1, 1 < class 1 ≤ 2, *etc.*). Age-specific growth trajectories were estimated from this dataset by fitting von Bertalanffy, Gompertz, and logistic growth models to the data (*Nelson, 2019*) (Eqs. (2)–(4), respectively):

$$TL_t = TL_{inf}\left[1 - e^{-K(t-t_0)}\right] \tag{2}$$

$$TL_t = TL_{inf} \cdot e^{-e^{[-G(t-t_0)]}} \tag{3}$$

$$TL_t = TL_{inf} \left/ \left[1 + e^{-G'(t-t_0)}\right]\right. \tag{4}$$

where $TL_t$ is *TL* at age *t*; $TL_{inf}$ is the fish asymptotic maximum length, *i.e.*, "the mean length that the fish would reach if they were to grow indefinitely" (*Froese & Binohlan, 2000*); in Eq. (2), *K* is a growth coefficient and $t_0$ is the theoretical age when *TL* = 0. In Eqs. (3) and (4), *G* and *G'* are growth-rate coefficients, and $t_0$ is time at the curve's inflection point (Gompertz and logistic models are sigmoid-shaped), when instantaneous growth rate is maximum (*Quist, Pegg & Devries, 2012*; *Tjørve & Tjørve, 2017*). In the Gompertz model, the absolute maximum growth rate at the curve's inflection point (*Tjørve & Tjørve, 2017*) was estimated as AMGR = $(G \cdot TL_{inf})/e$, where *G* is the Gompertz growth-rate coefficient and $e = 2.71828$ is Euler's number. The best model was selected based on the Akaike information criterion corrected for small samples (AICc) ("Statistical analyses"). Model

assumptions of homoskedasticity and normality of residuals of the selected model were evaluated using a residual plot and a QQ plot, respectively. Because of the absence of individuals in age classes 7 and 8 in the dataset, and the presence of only three individuals in age classes 9 and 10, the effect of the heterogeneous distribution of available length-age data on the growth-trajectory model was evaluated by performing a sensitivity test, *i.e.*, by fitting the selected growth-trajectory function without the three larger individuals.

The mean length at first maturity ($TL_m$) was estimated using the $TL_{inf}$ estimated by the selected growth model, according to the empirical relationship (*Froese & Binohlan, 2000*):

$$\log_{10}(TL_m) = 0.8979 \cdot \log_{10}(TL_{\text{inf}}) - 0.0782 \tag{5}$$

Optimum length, or the length at maximum yield per recruit, was defined as the length at which "the product of the number of surviving individuals multiplied by their average weight results in the highest biomass" ($TL_{opt}$; *Froese & Binohlan, 2000*). $TL_{opt}$ is typically larger than $TL_m$, and often corresponds to the maximum female fecundity (*Beverton, 1992*; *Froese, 2004*). *Froese (2004)* identified three key risks of fishery management: (i) recruitment-overfishing, *i.e.*, decreasing recruitment by catching fish before their first reproduction, (ii) growth-overfishing, *i.e.*, catching fish before they can realize their growth potential, and (iii) reducing the number of 'mega-spawners', *i.e.*, fish that disproportionally contribute to recruitment due to their large size, thus increasing the resilience of the population to random recruitment fluctuations. In order to mitigate these risks, *Froese (2004)* argued that the whole catch should only contain fish with $TL = TL_{opt} \pm 0.1\ TL_{opt}$. Optimum length was estimated using the estimate of $TL_{inf}$ obtained from the selected growth model, according to the empirical relationship (*Froese & Binohlan, 2000*):

$$\log_{10}(TL_{opt}) = 1.0421 \cdot \log_{10}(TL_{\text{inf}}) - 0.2742 \tag{6}$$

The same dataset used to estimate age-specific growth trajectories ($n = 295$; Table S1; Fig. 2A) was used to estimate the population's instantaneous total mortality rate ($Z$), using an unbiased Chapman–Robson estimator ($Z_{CR}$; *Chapman & Robson, 1960*; *Hoenig, Lawing & Hoenig, 1983*; *Smith et al., 2012*; *Ogle, Wheeler & Dinno, 2021*; "Statistical analyses"). Assuming that the fish catch is proportional to the size of the population, the total annual mortality rate was estimated from a catch curve obtained from cross-sectional data, plotting catch numbers of 2016–2020 against fish age class (catch-at-age). Vectors of cross-sectional data are equal to vectors of longitudinal data, when assuming that (i) the mortality rate is constant across time and among ages (constant mortality), (ii) recruitment is constant for each cohort or fluctuates randomly (constant or randomly fluctuating recruitment), and (iii) the probability of capture is constant across time and among ages (constant vulnerability). Catch-curve methods further assume that (i) the population is closed (no outputs except mortality and no inputs except recruitment), (ii) $Z$ is constant across ages and years on the descending limb of the curve, and (iii) the sample is unbiased for any age class (*Ogle, 2019*; *Ogle, Wheeler & Dinno, 2021*). Within the limits of these assumptions, the Chapman–Robson method estimates the total annual survival rate $S$ along the descending limb of the catch curve, not including the first fully-recruited age (peak of the curve, in our sample: age 2; Fig. 2A), *i.e.*, including age classes 3–6 ($n = 80$;

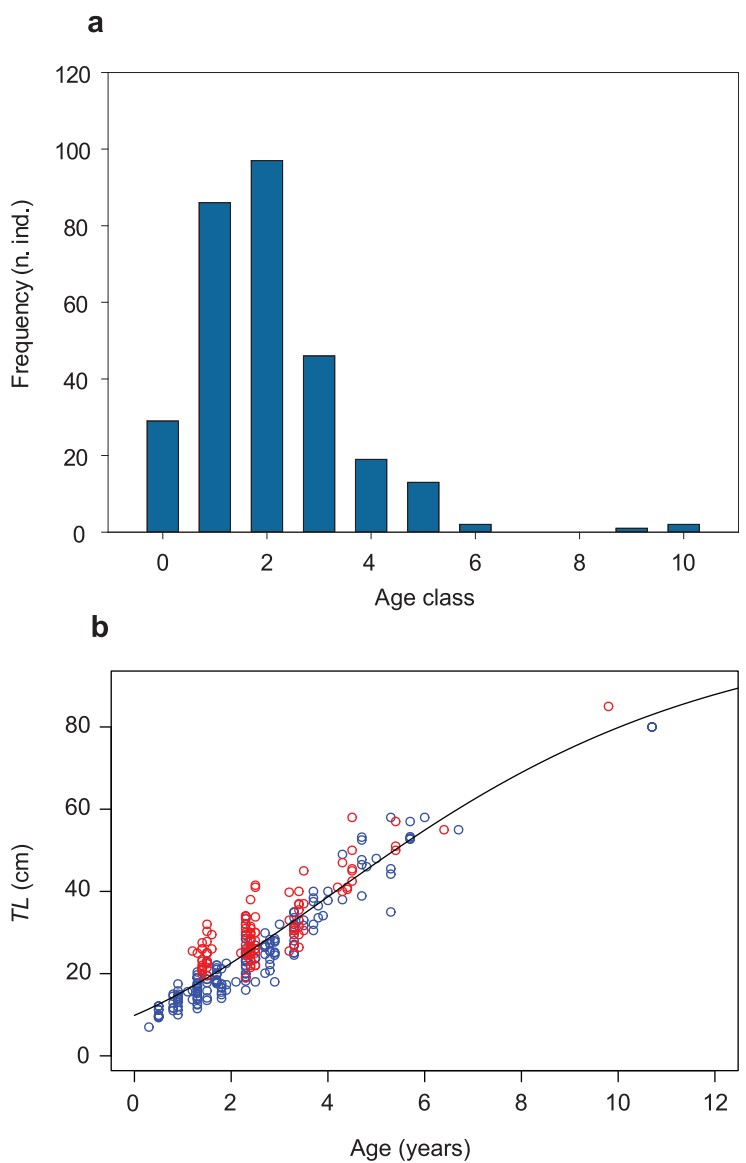

**Figure 2 Age-class frequency distribution, length-age relationship and growth trajectory shown for the population of *S. marmoratus* of the Toce River (*n* = 295).** (A) Age-class frequency distribution; n. ind.: number of individuals; each age class corresponds to a time interval of 1 year. (B) Length-age relationship with fitted Gompertz model (black curve); blue circles: sample A; red circles: sample B; *TL* (cm): total length.

Fig. 2A). *S* is then used to obtain an unbiased estimate of instantaneous total mortality rate, *i.e.*, $Z_{CR}$.

$Z_{CR}$ values were then substituted for $Z$ to calculate annual total mortality rates ($A_Z$) as (*Ogle, 2019*):

$$A_Z = 1 - e^{-Z} \tag{7}$$

The instantaneous natural mortality rate ($M$) of the population was estimated by using the maximum recorded age of *S. marmoratus* ($t_{max}$ = 11+ years, northern Italy; *Jelli, Alessio*

& Duchi, 1996), using the empirical one-parameter $t_{max}$-based estimator (*Then et al., 2015*; "Statistical analyses"), as:

$$M = 5.109/t_{max} \tag{8}$$

The proportion of the population that suffers natural mortality in 1 year ($A_M$) was obtained from the $M$ estimate as in Eq. (7).

The instantaneous fishing mortality rate ($F$) of the population was calculated as (*e.g.*, *Ogle, 2019*):

$$F = Z - M \tag{9}$$

The proportion of the population that suffers fishing mortality in 1 year ($A_F$) was obtained from $F$ as in Eq. (7). The exploitation ratio $E$, *i.e.*, the proportion of recruits that is fished during all the years of their life (*Ricker, 1975*) was computed as:

$$E = F/Z \tag{10}$$

Assuming that fishing and natural mortality rates follow a constant or parallel yearly variability, $E$ roughly assesses whether a stock is overexploited, based on the assumption that the optimal value of $E$ ($E_{opt}$) is approximately 0.5, *i.e.*, that sustainable yield is optimized when $F = M$ (*Gulland, 1971*).

## Statistical analyses

Several packages of R statistical software (v.4.0.5; *R Core Team, 2021*) were used. Length-weight relationships were analyzed using the 'moderndive' R package (*Kim, Ismay & Bray, 2018*). Von Bertalanffy, Gompertz, and Logistic age-specific growth models were fit to length-at-age data with R packages 'FSA' (*Ogle, Wheeler & Dinno, 2021*), 'nstools' (*Baty et al., 2015*), and 'fishmethods' (*Nelson, 2019*), using initial values from *Loro & Zanetti (1996)*, to provide a rough fit to the data (*Ogle, 2019*). Basic data elaborations were conducted with 'dplyr' (*Wickham et al., 2021a*), and graphs were prepared with 'ggplot2' (*Wickham, 2016*), 'FSA', and 'svglite' (*Wickham et al., 2021b*). Age-specific growth models were compared using the Akaike information criterion corrected for small sample size, with the 'AICcmodavg' R package (*Mazerolle, 2020*). The FSA package was also used to obtain an unbiased estimate of instantaneous total mortality rate ($Z_{CR}$), and the instantaneous natural mortality rate ($M$). The 95% confidence interval (CI) of model parameters '$P$' ($P = TL_{inf}$, $t_0$, $G$ or $K$ or $G'$) were computed as $P \pm c\cdot$s.e., using their s.e., df values, and 97.5[th] percentiles of the t-distribution ($c$). For $TL_m$ and $TL_{opt}$, we estimated the 95% prediction interval (PI) of the population mean lengths as:

$$10^{\left[\left(a\cdot\log_{10} TL_x - b\right)\pm c\cdot \text{s.e.}\right]} \tag{11}$$

where $a$ and $b$ are the slope and the intercept of the empirical relationships (5) and (6), df = n−2, and $TL_x$ is $TL_m$ or $TL_{opt}$, assuming that (i) the published s.e. values in *Froese & Binohlan (2000)* are computed by first dividing the sum of squared residuals with sample size less two, and then taking square root; and (ii) the parameters of interest in the 95% CI

are the population means of '$L_m$' and '$L_{opt}$', respectively (*Froese & Binohlan, 2000*). We used a 5% level of significance in all statistical tests.

## RESULTS

The total sample includes 579 trout specimens with marbled coloration elements, likely including 'genetically pure' S. marmoratus, S. trutta × S. marmoratus hybrids, and variably introgressed individuals (samples A, $n = 466$; sample B, $n = 113$; Tables 1, S1; Fig. S1). Fish are significantly larger and older in the middle tract of the Toce River (*TOR2*; Fig. 3), however, large fish (≥80 cm *TL*) were found in all river sections (Table 1), and the local length record was caught in the upper Toce River, near the confluence with the Diveria Torrent (*TOR3*; Fig. S3). The measured numerical and biomass densities are higher in *TOR2* than in *TOR1* and *TOR3*, though not significantly so (Kruskal–Wallis test, $p = 0.299$ and $p = 0.081$, respectively; Fig. 4). Six individuals collected in the middle Toce river and the individual collected in Lake Maggiore (21.5–85 cm *TL*, 1.5–9.8 years) exhibited one or more traits compatible with convergent traits found in trout living in deep waters and pelagic habitats (*e.g.*, *Snoj et al., 2010*), *i.e.*, contracted marbled coloration patterns, silvery background color, pointed caudal-fin lobes ('swallow-tail'), and intact ventral caudal lobes (*e.g.*, Fig. S4).

The estimated $\log_{10}$ length (*TL*) to $\log_{10}$ body mass (*W*) relationship is (Table 2, Fig. S5):

$$\log_{10}(W) = 3.055 \cdot \log_{10}(TL) - 2.100 \tag{12}$$

The Gompertz growth function has the lowest AICc value among the models tested (Fig. S6; Tables S3, S4). A biologically unrealistic $TL_{inf}$ was estimated using the Von Bertalanffy model (Table S3; Fig. S6). The $TL_{inf} \sim 112$ cm estimated without the three larger individuals (sensitivity test; Table S5; Fig. S7) is not larger than 10% of that one obtained including them ($TL_{inf} \sim 105$ cm, $G \sim 0.2$, and $t_0 \sim 4$ years; Table 2; Figs. 2B; S8), indicating that the presence of a gap in the dataset of age classes 7 and 8 and the presence of only three individuals in classes 9 and 10 did not significantly bias the analysis. The growth trajectory fits a sigmoid-shaped model, suggesting a gradually accelerating growth rate at earlier life-history stages, and a slight deceleration after an average age ($t_0$) of ~4 years (*Quist, Pegg & Devries, 2012*). The estimated AMGR is 8.33 cm year$^{-1}$.

The mean length at sexual maturity is $TL_m \sim 55$ cm and the optimum length is $TL_{opt} \sim 68$ cm (Table 2). The unbiased estimate of instantaneous total mortality rate is $Z_{CR} \sim 0.9$ and the total annual total mortality rate is $A_Z \sim 0.6$; *i.e.*, an average of 60% of the population die annually (Table 2). The empirically estimated natural instantaneous mortality rate is $M = 0.464$. The total annual natural mortality rate is $A_M = 0.372$, *i.e.*, ~40% of the population die naturally each year. The fishing instantaneous mortality rate is $F = 0.460$ and the total annual fishing mortality rate is $A_F = 0.369$, that is ~37% of the population are fished annually. The exploitation ratio is $E = 0.498$, indicating an optimized sustainable yield.

However, following the rationale by *Froese (2004*; $TL_{opt} \pm 0.1\ TL_{opt}$), the total catch should only include fishes ranging ~61–75 cm *TL*, in order to prevent overfishing and increase recruitment resilience. The frequency distribution of 1-cm *TL* classes of our total

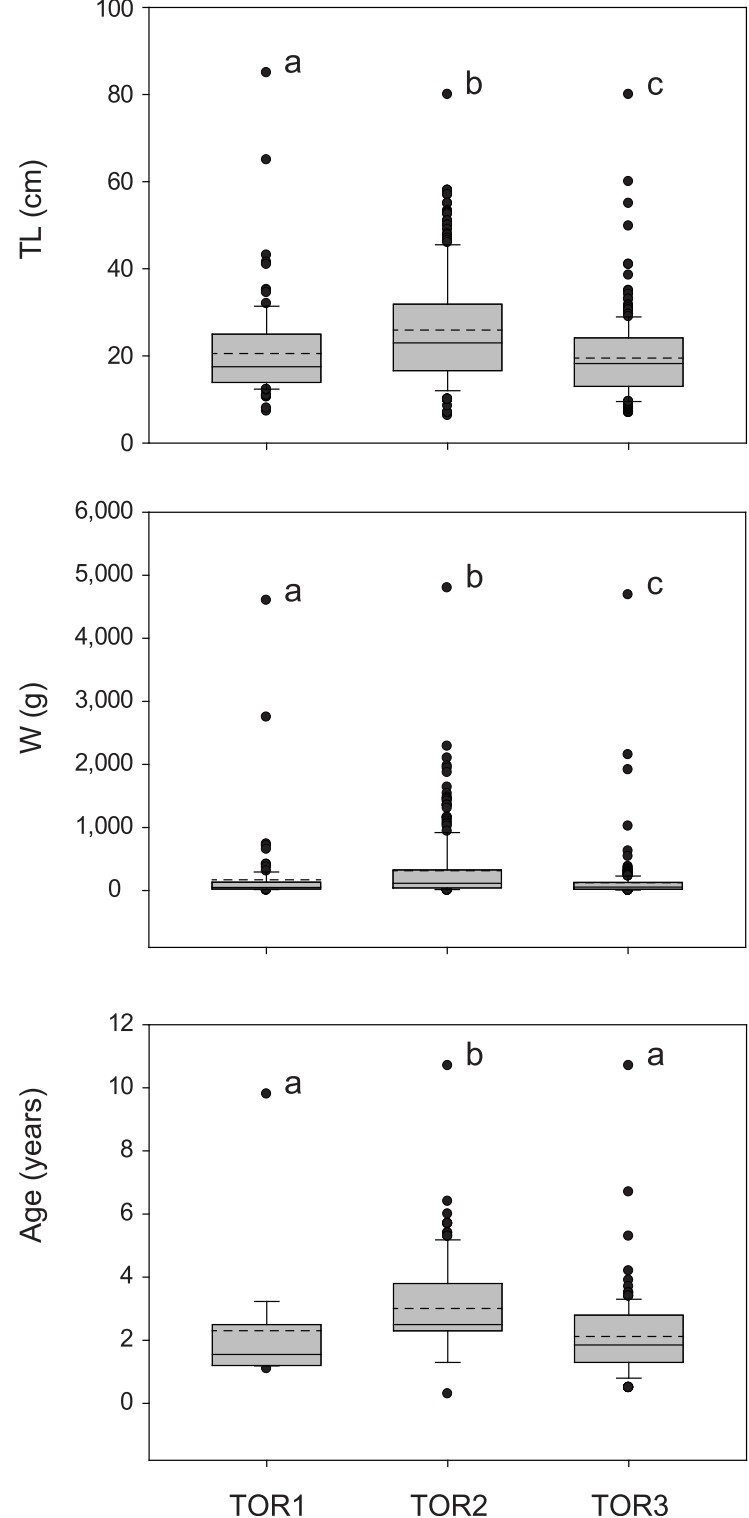

**Figure 3 Distribution of size (total length *TL*, wet mass *W*, *n* = 579; Table 1) and age (*TL-Age*, *n* = 295, Table 1) of *S. marmoratus* in the three sampled tracts of the Toce River (lower *TOR1*, middle *TOR2*, and upper *TOR3*).** Boxplots: dashed horizontal line: mean value; solid horizontal line:

**Figure 3** (continued)
median; box boundaries: 25th and 75th percentiles; whiskers: 90th and 10th percentiles; black dots: outliers; different letters above boxplots indicate significant statistical difference between river tracts. The *p*-values were computed from the Kruskal–Wallis test and *post-hoc* Mann–Whitney pairwise tests with Bonferroni correction.

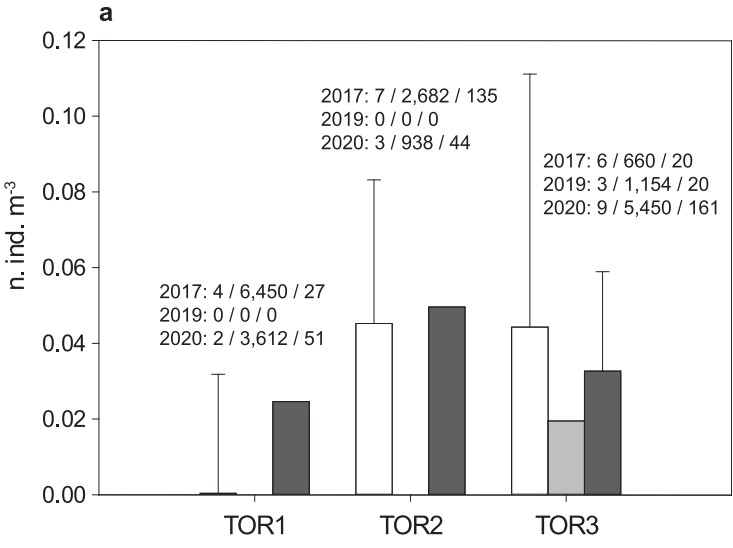

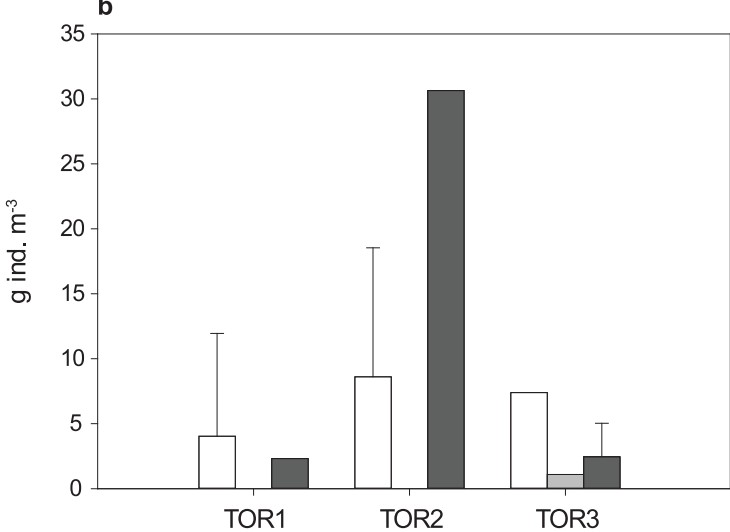

**Figure 4 Density distribution of *S. marmoratus* in 2017 (white bars), 2019 (light gray bars), and 2020 (dark gray bars).** Density was estimated using georeferenced electrofishing transects along the lower, middle, and upper tracts of the Toce River (*TOR1* to *TOR3*; Fig. 1). (A) Numerical density (number of individuals m$^{-3}$); above bars, for each year: number of transects/sampled water volumes in m$^3$/number of individuals; grand mean density per river tract throughout the years: 3 ± 2 s.d. n. ind./100 m$^{-3}$. (B) Biomass density (g m$^{-3}$); grand mean density per river tract throughout the years: 8.08 ± 10.33 s.d. g m$^{-3}$; bars: mean density per river tract +1 s.d. (whiskers; s.d. was estimated when the number of transects is ≥4).

**Table 2  Estimates of the parameters of the length-weight regression, Gompertz growth, and Chapman–Robson exponential mortality models.**

|  | TL-W intercept | TL-W slope | $TL_{inf}$ (cm) | G | $t_0$ (years) | $Z_{CR}$ | $A_Z$ | $TL_m$ (cm) | $TL_{opt}$ (cm) |
|---|---|---|---|---|---|---|---|---|---|
| Value | −2.100 | 3.055 | 105.0 | 0.216 | 3.99 | 0.924 | 0.603 | 54.529 | 67.933 |
| s.e. | 0.022 | 0.016 | 7.509 | 0.016 | 0.358 | 0.141 | n.a. | 0.127[*] | 0.073[*] |
| CI/PI | (−2.143, −2.058) | (3.024, 3.087) | (90.2, 119.7) | (0.184, 0.248) | (3.29, 4.69) | (0.648, 1.201) | (0.477, 0.699) | (30.7, 96.8)[§] | (48.8, 94.6)[§] |

**Notes:**
[*] Value from *Froese & Binohlan (2000)*.
[§] Prediction Interval.
s.e., standard error of parameter estimate; CI/PI, 95% confidence interval or prediction Interval; n.a., not available.

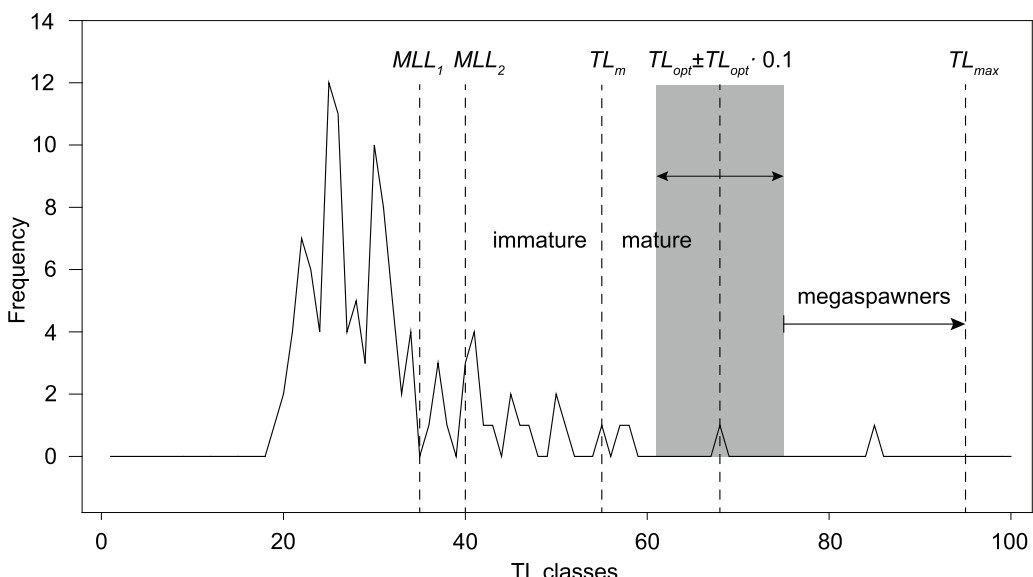

**Figure 5  Length-frequency distribution of angling catches of *S. marmoratus* in the Toce River and Lake Maggiore (sample B, *n* = 113).** *MLL*, minimum-length limit for retaining captured *S. marmoratus*; $MLL_1$, minimum-length limit for *S. marmoratus* in the VCO Province before 2019 (35 cm *TL*); $MLL_2$, minimum-length limit after 2019 (40 cm *TL*). $TL_m$, estimate of mean length at first maturity; $TL_{opt}$, estimate of optimum length; $TL_{opt} \pm 0.1\ TL_{opt}$, length interval proposed by *Froese (2004)* to prevent overfishing risks, *i.e.*, removing reproductive individuals and individuals which do not provide the maximum yield, and removing large fish disproportionally contributing to recruitment (megaspawners); $TL_{max}$, length record of *S. marmoratus* in the Toce River.

angling catch (sample B, *n* = 113; Table 1; Fig. 5) shows that 16–23% of the total catch would have been retained, according to the past and current regulations (minimum length limit *MML* for *S. marmoratus* in the VCO Province before and after 2019). Only one fish (4–6% of the retained catch) has a size (68 cm *TL*) within the optimum length interval proposed by *Froese (2004)*. Outside this interval, only two retained fish (8–11%) are mature individuals that could have reproduced before capture, while 14–22 retained fish (78–85%) are immature individuals that likely never reproduced. The only captured megaspawner (85 cm *TL*) would have been retained (Fig. 5).

## DISCUSSION

The marble trout is a species of high conservation interest (*Bianco et al., 2013*). Despite being widely distributed in the southern Alpine region, where it is typically found in large

subalpine rivers and tributaries, the published information on the structure of these populations is fragmentary.

Some comparisons with published data on population density can be made. The Toce River marble trout population is 4–7 times denser in terms of number of individuals and more variable in terms of biomass (from five times less dense to 50 times denser in different sites) than that of the middle tract (15 km) of the Brenta River (Southeastern Alps; *Turin, 2000*). The Slovenian populations (Southeastern Alps) live in small and fast-flowing karstic headwaters, and are characterized by individuals with smaller average size, shorter lifespan, and earlier sexual maturity. Relative to them, the Toce River population is on average less dense in terms of number of individuals, but denser in terms of biomass (*Šumer, Leiner & Povž, 2001*). The higher average density, significantly larger size, and older age of the trout collected in the middle tract of the Toce River might indicate that this system offers more optimal conditions for growth, or might suggest the presence of ontogenetic migration patterns among the different river tracts, *e.g.*, a tendency of moving to the middle river tract during growth. However, it should be noted that the relatively low density in the lower tract of the Toce River might be an effect of lower electrofishing and rod-and-line sampling efficiency in deep waters. In order to avoid similar sampling biases potentially caused by the low sampling efficiency in deeper waters, some previous studies explicitly limited populations studies to selected smaller and shallower tributaries of large rivers (*e.g.*, *Crivelli et al., 2000*).

Log-log regressions between length and body mass are largely consistent with those previously estimated for the Brenta River population (*Turin, 2000*), and for both Slovenian and northern Italian populations (*Lorenzoni et al., 2012*). The growth trajectory model estimates a maximum growth rate of 8.2 cm $decy^{-1}$ at 3–5 years (inflection point), consistent with their dietary shift to a piscivory diet, at ~40 cm *TL* (*Turin, 2000*). At 3–5 years of age, the estimated fish sizes are consistent with those of the High-Po (Southwestern Alps), Adige, Piave, Avisio (*Jelli, Alessio & Duchi, 1996*) and Brenta Rivers' populations (Central-eastern and Southeastern Alps; *Turin, 2000*). The asymptotic length estimated by the selected growth model is biologically compatible with maximum size records (139 cm *TL*~ 120 cm standard length (*SL*), Rienz River (*Delling, 2002*); *SL*/ *TL* = 0.864 ± 0.015, $n$ = 14). However, comparisons with growth models based on different populations should be avoided, since estimated parameters depend on local ecological conditions, such as temperature, food availability, current patterns, and local phenotypic adaptation (*e.g.*, *Vincenzi et al., 2007a*).

Although the exploitation ratio suggests that the Toce River population is sustainably exploited, $F = 0.91M$ is from 5% to 36% larger than optimal fishing mortalities estimated for teleosts ($F_{opt}$ = 0.67–0.87; *Zhou et al., 2012*; *Lester et al., 2014*). Following the rationale by *Froese (2004)*, the length-frequency distribution of our angling sample and estimated $TL_{opt}$ and $TL_m$ suggest that the present regulations (*MLL*) exacerbate risks of overfishing and loss in resilience to recruitment fluctuations. Most of the captured fish were also immature individuals, which may be either wild-born offspring or stocked fish. However, the length frequency distribution also shows a conspicuous lack of mature individuals larger than *MLL*, suggesting a significant effect of fishing pressure on the population

structure. Resilience to recruitment fluctuations might be improved by implementing new regulations allowing to retain only fish of ~60–70 cm *TL*, thus both preventing overfishing risks and preserving 'mega-spawners'. Harvesting regulations based on length interval limits are referred to as 'harvest-slot length limits' (*HS*). In the observed scenario, *HS* regulations would likely promote more balanced population size structures, higher biomass, density, and fecundity (*Ayllón et al., 2019*), and are expected to gradually increase the frequency of larger reproductive individuals in the population. However, changing the present *MLL* regulations to the proposed *HS* (60–70 cm *TL*) would have the immediate effect of drastically decreasing the harvestable catch. In our angling sample, this would correspond to a decrease from ~19% (*n* = 21) to ~1% (*n* = 1) of the total catch. Therefore, these actions could be only implemented with the empathetic support of the anglers and sporting fishing associations.

No genetically 'pure' populations have been found in northern Italy, contrary to those living in the drastically different Slovenian karstic ecosystems (*e.g.*, *Šumer, Leiner & Povž, 2001*). Studies that phenotypically discriminated between 'pure' and hybrid individuals (*e.g.*, *Turin, 2000*) are unreliable, as previously explained. On the other hand, sampling an introgressed marble trout population by selecting individuals with marbled elements in their coloration patterns may leave out hybrids or variably introgressed individuals without marbled coloration elements. No study in this region described the size-age structure of a marble trout population by analyzing the individual level of genetic introgression. These limitations of this and previous studies should be addressed by future investigations.

The combined effects of climate change and anthropogenic genetic introgression might challenge the sustainable management of marble trout populations, in order to face the contrasting goals of (i) favoring predicted upriver migrations induced by global warming, *e.g.*, by creating ecological corridors, and (ii) preventing genetic mixing between subpopulations with different levels of genetic introgression presently separated by artificial hydrological barriers (*e.g.*, *Splendiani et al., 2019*). Furthermore, although there is evidence that pure and hybrid individuals might have distinct ecological and physiological traits (*e.g.*, *Meraner et al., 2010*; *Simčič, Jenšenšec & Brancelj, 2015a*, *2015b*, *2017*), such studies were either conducted on artificial populations, or on extremely inbred and small populations living in karstic headwaters, and are also not comparable with our results.

Typically admixed and sympatric ecophenotypes with different life histories, not uncommon in salmonids (*e.g.*, *Charles et al., 2005*; *Snoj et al., 2010*), could have divergent ecology and behavior, with distinct population size-age structures. The presence of some individuals exhibiting traits compatible with pelagic morphs might suggest the presence of different ecophenotypes in the Toce River, such as lacustrine, fluvial-adfluvial, or lacustrine-adfluvial morphs. However, further genetic and ecomorphological studies are needed to confirm this hypothesis, since the only available study lacks experimental evidence of migratory patterns (*Gibertoni et al., 2014*).

## CONCLUSIONS

The description of density distribution, length-weight relationship, and length-age structure, and the estimates of growth trajectory, mean length at first maturity, optimum length, and mortality provide a first overview of the structure and dynamics of the Toce River marble trout introgressed population. The overlap of these estimates on the length-frequency distribution of our angling sample highlights the presence of overfishing risks posed by the current fishing regulations.

Future research may focus on (i) conducting genetic analyses on all sampled individuals, thus evaluating the effects of genetic introgression on population size-age structure; (ii) improving the sampling design to obtain more spatiotemporally balanced datasets, *e.g.*, sampling periodically in a smaller number of selected sites; (iii) using river sections that allow a higher and more homogeneous fishing efficiency in shallow waters, as a genetic and ecological proxy of the larger system; (iv) tracking fish movements in the river and lake systems, investigating possible ontogenetic shifts, individual patterns, and presence of ecophenotypes with different behaviors and ecology; and (v) quantifying the relative effects of harvesting larger fish and stocking smaller fish on the population structure. In this latter case, measurements of the genetic distance between smaller fish in the wild population and in hatchery stocks might preliminarily demonstrate the actual contribution of stocking to the wild river population.

The long-term sustainability of inland recreational fisheries is a priority, if shared economic, research, and conservation-oriented goals are to be achieved (*Cooke et al., 2015*). Management strategies of such socio-ecological systems must merge the priorities and needs of all the stakeholders, from conservation scientists to anglers, if the common goal of bequeathing the structure and function of these valuable ecosystems to future generations is to be achieved.

## ACKNOWLEDGEMENTS

The authors thank Jordi René Mor and Stefano Brignone for their assistance in the field, Massimiliano Ghibaudo for his photos of the marble-trout length record for the Toce River, Simone Sechini for mathematical elaborations of the Gompertz model, and Michela Rogora (Water Research Institute, IRSA-CNR) for the data from the CNR Candoglia weather station. Thanks also to an anonymous reviewer and Amy Then Yee Hui (Institute of Biological Sciences, Universiti Malaya) for their constructive comments and criticisms during the revision of earlier versions of this manuscript.

### Funding

This study was funded by the grants LIFE15 NAT/IT/000823 IdroLIFE, Interreg ITA/CH SHARESALMO, and Fondazione Cariplo "Dal Mare all'Orta". The funders had no role in study design, data collection and analysis, decision to publish, or preparation of the manuscript.

## Grant Disclosures

The following grant information was disclosed by the authors:
LIFE15 NAT/IT/000823 IdroLIFE.
Interreg ITA/CH SHARESALMO.
Fondazione Cariplo "Dal Mare all'Orta".

## Competing Interests

The authors declare that they have no competing interests.

## Author Contributions

- Gianluca Polgar conceived and designed the experiments, analyzed the data, prepared figures and/or tables, authored or reviewed drafts of the article, and approved the final draft.
- Mattia Iaia performed the experiments, prepared figures and/or tables, and approved the final draft.
- Paolo Sala performed the experiments, prepared figures and/or tables, and approved the final draft.
- Tsung Fei Khang analyzed the data, authored or reviewed drafts of the article, and approved the final draft.
- Silvia Galafassi analyzed the data, prepared figures and/or tables, and approved the final draft.
- Silvia Zaupa analyzed the data, prepared figures and/or tables, and approved the final draft.
- Pietro Volta conceived and designed the experiments, analyzed the data, authored or reviewed drafts of the article, and approved the final draft.

## Data Availability

  The raw data are available in the Supplemental File.

## Supplemental Information

Supplemental information for this article can be found online at http://dx.doi.org/10.7717/peerj.14991#supplemental-information.

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
