# Peer review of "Size-age population structure of an endangered and anthropogenically introgressed northern Adriatic population of marble trout (Salmo marmoratus Cuv.): insights for its conservation and sustainable exploitation"

_PeerJ, doi:10.7717/peerj.14991_

## Round 0.1 · original submission · Major Revisions

Both reviewers appreciate the work you've put into this paper and think it is a potentially valuable contribution, but have included a number of suggestions that will improve the manuscript.

Reviewer 1 ·

Basic reporting

Manuscript “Population structure of an endangered northern Adriatic population of marble trout (Salmo marmoratus Cuv.): Insights for its conservation and sustainable exploitation” describes size and age structure of the marble trout, including other allochtonous trout present in Toce river basin. The language used is clear, professional English. Introduction is clearly written, but some important references, describing similar studies on marble trout are missing (work coordinated by Crivelli, for instance: Vincenzi et al., 2007, 2008, 2010). Figures are relevant, but could be improved for clarity and distinction between studied taxa. Raw data are supplied.

Experimental design

Research is within the scope of the journal, but the aim of the study is not well addressed. Authors attempt to provide “baseline data for the conservation and management of the marble trout population of the Toce River system, as a case study exemplifying the possible effects of recreational fishery management.”, but without data on the number of annually catch or data about marble trout structure before massive recreational fishery. I suggest authors to rewrite the research question to fit the data analysed. As written above additional literature about population structure of marble trout elsewhere should be included. Now it seems that this is the first study on population structure of marble trout.
Methods and also Results are hard to follow due to numerous abbreviations and numbers. I suggest authors to simplify their analyses, to present more results in graphs and not to have too many numbers in the text. They should appear in Tables, and should only be commented in the text.

Validity of the findings

Discussion is a bit weak, first stating results and tem trying to propose management options for marble trout. I think that this later is a bit too ambitious considering the type of the data analysed, but including data from fisheries catch. The same goes to Conclusions.

Additional comments

Beside complicated methods, multiple abbreviations used, the most weak part of the manuscript is a lack of clear definition which data were used in a particular analyses-marble trout or all the trout. This is obvious already in the Abstract, where it is not clear which data are meant (see l. 25-28). I suggest the authors to do the analyses separately for marble and brown trout. Being one of the largest salmonids, S. marmoratus is expected to be larger than S. trutta and have different growth rate. The data for the two species should be therefore presented separately (all the data included in the manuscript).
I suggest authors to completely rewrite the manuscript from Material and methods to Conclusion, to make it easier to follow and to comment and make conclusions based on the data analysed and results presented. I also suggest that the authors try to avoid mentioning the numbers and repeating results in discussion and conclusion.

l. 30: please, define what is “stock-length” and “quality-length”,
l. 96: have you analysed data separately for each year? It would be interesting to see, if the population structure changes throughout the analysed years.
l. 98 and 132: please, add the text describing the removal of scales from marble trout. Were the fish narcotised when handling them?
l. 117: it is not possible to detect and distinguish hybrid taxa with such technique and this might have a great effect on your results, when hybridization is present. Please, add description about phenotypical characterisation and distinction between S. trutta and S. ghigii.
l. 123: subsets of samples should somehow be marked in the Table. It is also not clear if you used samples from different time points to assess the biomass? If yes, please add this information and also explain, how this could impact the assessment. Have you controlled for the re-catches?
l. 351-353: “…more reasonable management of the Toce River marble trout recreational fishery”. More reasonable than what? Please, add information.
l. 363: “suggesting a possible impact on the reproductive stock”. Impact of what? Please, add information.
l. 399: section 4.2. does not exist, also I can not find the three options in discussion
Figure 4: why average length was used and not individual data?
Supplemental figures were in *.psd format that is not supported in the authors instructions and could not be opened with basic Microsoft package. I can therefore not make comments of these supplemental data.

·

Basic reporting

In general the paper is reasonably well written to provide context for understanding the issues faced by the marble trout and some rationale for examining the population structure. The use of English is also reasonable for the most parts but will benefit from proofreading. More importantly, there are various parts within the paper that appeared disjointed and are missing important details and justification (see more below and in general comments). The discussion and conclusion would benefit from restructuring (see general comments).

The introduction, as it is, is leaning heavily on discussion of various threats but very limited links of these threats to population structure, which is the heart of this paper. Some background on the effects of overfishing or climate change or introduction of non native species on population structure (ideally of marble trout), e.g. decreasing mean length, is needed.

It would be important to provide context for the practice of sport fishing in the Toce River which is the focal area for studying the species. Since it is meant to be a case study ‘exemplifying possible effects of recreational fishery management’, what is the current fisheries management in place (if any), e.g. catch and release of small individuals, closed seasons? L.353 in discussion also mentioned about ‘more reasonable management’ but in comparison to what? I see that MLL is being used (L.361) but this should be mentioned much earlier.

Authors mentioned about comparison to previously described systems (L.85) and this implies that there are similar population structure studies for the species elsewhere – it is important to review available estimates for the species elsewhere in the introduction.

L. 40-43 It would be ideal to expand the discussion on the issues of unsustainable management and invasive species here specifically addressing sport fishing of salmonids. For example how widespread is the issue of overharvesting of salmonids and the use of hatchery-raised animals to restock rivers?
L.48-49 Sentence needs to be rephrased for accuracy, e.g. something along the lines of ‘The taxonomic distinctiveness of this species relative to other similar looking salmonids is consistent with…’ It would be helpful to provide some context for this sentence, and to combine it with the issue of hybridization and genetic introgression raised further along in the text
L. 64 It should be the ‘Italian marble trout’ instead of the ‘Italian IUCN’.
L. 68 Elaborate how climate change resulted in ‘connectivity loss, bottlenecks’ and ‘increased vulnerability to climate warming’.
L.70 Discussion on introduction of non-native species should be grouped together (mentioned earlier in L.58)

Table quality is acceptable but the figures need to be improved considerably to be useful (see general comments). Figure captions are choppy and segmented – need to be revised for clarity, e.g. Figure 2. a) Length-age relationship and b) age class frequency shown for the Toce River….

Experimental design

The section on Materials and Methods could be better structured with added details to improve understanding of the experimental sampling and analysis.
L. 98 Were the two sampling methods used concurrently (within the same spatial period)? The nature of the ‘methodological sampling biases’ should be described to explain why the two methods would be complementary, e.g. line fishing captures bigger animals compared to electrofishing that targets shallower waters where smaller animals use?

L. 117 What is the context of the ‘species taxonomic discrimination based on phenotypic traits’? While I understand that there is the issue of hybridization introgression, it was not stated how this affects the morphology of the animals and how this might impact the results, e.g. of numbers and growth estimates? This is briefly mentioned in discussion (L.324 and again hinted in L.351) but should be introduced much earlier. L.330 suggests that authors treated all S. marmoratus (introgressed or not) as a single ‘species unit’? Please be specific about this within the methods.

L. 129 I have a hard time understanding the purpose of imputing missing data for body-mass density estimates – what exactly is this used for? If marble trout is the focus, why are there measurements for S. trutta? Also what is the regression model equation and how is this different from equation 1? I see that data for these other species (S. trutta, S. ghigii, O. mykiss) are included in Tables 1 and 2 and they are very interesting in the context of examining inter-species differences but not relevant to the stated objective of the paper?

L. 132 If the subset of samples A and B already have age data (and how are they derived?), why use dorsolateral scales to determine their ages? Has this method been used for marble trout and validated? Additionally why age classes of 0.9 decimal years? I had assumed that the authors fitted the growth models to the raw (individual) length-age data but Fig 4(a) suggests that they were fitted to average length at age? If the former, the plot should show the individual lengths-at-age to show the variability in individual growth. If the latter, the authors should explain why this was done.

L. 148-151 Move Gompertz model description to line 146 before talking about model selection.

L. 152 onwards Why is this additional method to estimate TLinfE necessary? The Froese & Binohlan method is clearly one that would be less superior to the VB, Gompertz and logistic models that are fitted directly to the data collected, especially when AIC will be used to select the best growth curve and associated growth parameters. I do not see the need or the logic for doing this.

L. 178 What ‘same’ dataset? The one with n = 295? There were various data subsets being described prior – be specific.

L. 178 The description of the Chapman Robson model is excessively long (especially when considering that there is barely any description of the growth models used or their assumptions). Limit the description to mention of the assumptions and provide the equation used.

L. 209 Likewise no need to include the details of the study in Then et al. (2015).

The section on PSD can be better organized for clarity:
L. 238 Would be more useful to mention what the indices represent when describing PSD in L. 227. What is ‘stock-length fish’?
It is unclear to me how the five length categories and the associated %, e.g. stock (S’, 20-26% TLmax) are derived? Are these based on inputs from anglers on desired size range?
L. 241 Based on Table 4, it seems that both TLmax and and TLmaxG are used to calculate incremental PSD values? The information presented in L.243-244 is without context? Is the intention to present a formal length-length relationship?
Additionally, it would be useful to provide some indicators on what the PSD values mean for the size structure.

L. 248 Please clarify what the examination of satellite images are for (and where the images are taken from)?

Validity of the findings

I commend the authors for their extensive data set and analysis. In general the analysis approaches are sound (see exceptions discussed earlier in the materials and methods section that need to be addressed to ensure validity of findings). However the multiple reported estimates in the Results section would be much easier to follow if presented in a Table format, especially for the different growth estimates and their associated AICc values. Please include 1-2 tables to summarize the results concisely. For ease of reading, I would also strongly recommend to drop the hyphen from the names of the tracts, e.g. TOR1 instead of TOR-1.

In relation to the growth estimates, I actually would like to see visual fits of the other growth models that were fitted as well. What is evident from Fig 4 is the limited number of larger/and older individuals. It is not surprising that the Gompertz one was selected but given that the authors compared their TLinf estimate to a Von Bertalanffy derived estimate (L.349) it seems that one important issue is undersampling of larger individuals, which is not discussed by the authors?

Figure 2. a. – should be length – age relationship (and appears to be the same as Fig 4 a – there is no need for two of the same figures, replace the current Fig. 2a with Fig. 4a. – Fig 4b and c unnecessary as they are showing visually if assumptions are met – which should be described in the method when discussing growth model selection). I would like to see the individual length-at-age plotted with indication of which animals came from either sampling methods (using two different symbols).

Figure 3. I assume that what is presented is the average density and biomass and if so, state this clearly. Given that GHIG and OMMY are barely visible, they can be omitted from the figure. What would be important is to show how variable the estimates of biomass and density are across the tracts (with indication of no of times sampling per tract and average depth).

L.320 Rephrase – it is the authors’ job to interpret the results. Regarding this bias, it would be important to provide data to show that there might be this bias, including providing average depth estimates for the middle and ‘high?’ (should be upper) tracts? (This is tied to my comment on Fig.3 in general comments). Provide supporting refs from the literature on this bias and how one could potentially address this for future studies.

I have not seen anything on ethical approval/declaration in using electrofishing for sampling in the paper?

Conclusions – should be a brief overview of the findings, linking to the objectives, and way forward. However most of the write up appeared to be much more suited within the discussion section, especially in developing the last discussion para on alternative management.

Additional comments

L. 17 Add ‘inland fisheries’
L. 18 Instead of unavailable (since the data are now available), a more appropriate word would be ‘novel’. Also ‘..age structure of (instead of one)’
L. 21 There is no real context for the mention of the LIFE15 NAT/IT/000823 project?
L. 123 I assume it should be fish density and biomass.
L.317 I would not use the word scarce given the available studies being compared to in L. 332 onwards.
L. 329 Authors appeared dismissive of evidence by Gibertoni et al. (2014) but with no clear context on why they might think otherwise, i.e. the fish do not migrate. Since authors sampled both the lakes and rivers, it would be useful to counter the evidence with their own observations, e.g. were there different phenotypic traits observed?
L. 341 A single sentence paragraph?
L. 354 What is highly variable in salmonids?
L. 356 MLL not MML.
L.358 What exactly is the number referring to in ‘annual controls of 3%’? Control of no of visitors?
L. 361 So MLL implies that you can keep animals >40 cm TL (not necessarily mature yet) – why is this an impact on reproductive stock? Rephrase
L. 363 ‘Furthermore’ instead of ‘Further’
L. 369 ‘Observation’ iii is not an observation but an assessment, and obs I and ii is supporting the assessment. One thing the authors have not acknowledged – the issue (limitation) with the estimated mean length at first maturity?
L. 371 Why would ‘relatively small and young mature individuals’ be ‘an effect of stocking activities’ – need to elaborate? Also, no context provided on stocking activities done in the lake/ river?
L. 377 I don’t see observation v?
L. 375 The paragraph in its current form is not well structured or developed in its arguments – the current impression is that the authors are not sure whether the fishery is overexploited or close to optimum yield. Please restructure. The paragraph should also be split to two – move the discussion on observation iv and lines 377 onwards about exploring alternative management as a separate para. Clarify what is meant by HS and why this would be ‘promoting a more balanced population size structure, higher biomass, density, and fecundity of older and larger fish’.
The authors talked about climate change effects in the introduction but nothing of this is mentioned in the discussion? What important climate resilient management could be considered?

---

## Round 0.2 · Minor Revisions

The reviewer appreciates the work you've put into this paper and thinks it is significantly improved. They have included a few minor comments that I think will improve the paper further.

Reviewer 1 ·

Basic reporting

Manuscript “Size-age population structure of an endangered and anthropogenically introgressed northern Adriatic population of marble trout (Salmo marmoratus Cuv.): Insights for its conservation and sustainable exploitation” describes size and age structure of the marble trout, present in Toce river basin.
Revised version is more focused and easier to follow. The aim is more clearly defined. There is however a problem with taxon determination and selection. Authors wrote that they selected only marble trout according to marbled elements. First, they should clearly define what is meant by the term “marbled elements”. Besides, they report that in Italy, most probably also in Toce river (also evident from river management description), there is no pure marble trout present. Although it is clearly noted in the manuscript title, that the study includes anthropogenically introgressed marble trout, this is not clear from the text, except for the beginning of Discussion. To my opinion, the fact that this study most probably includes hybrid individuals that are phenotypically determined as marble trout, should be stressed in the Introduction, aims and also in Methods and Results (see further comments). The study would much benefit if genetic data of these individuals would be also presented. Authors cite their submitted manuscript (Polgar et al., submitted) where this data are available and I really think it's a pity that they didn't present these two data sets together.
The language used is clear, professional English. Introduction is clearly written, including relevant references. A part of Introduction (lines 131 to 164) to my opinion does not fit in this section, but should rather be included as a separate section describing the Toce river management, within Material and methods or as a supplemental data. Figures are relevant and raw data are supplied.

Experimental design

Research is within the scope of the journal, and research question is defined. Methods are appropriate, described with sufficient information and more clarity in revised version. I would only like to point to the problem of marble trout selection, described above. Please, consider this note and pay more attention to use more appropriate description of fish studied.

Validity of the findings

All underlying data have been provided; they are robust, statistically sound and controlled.
Discussion and conclusion of revised version are clear, following the original aims and presented results. I would suggest the authors to reorganize the discussion, first mention your main results and then the limitations of the study, and discuss the results considering the limitations. Now, the limitations are mentioned first, which leaves the impression that the research was wrongly designed and that some other studies should be done before this one (which is true on one hand and it is good that the authors are aware of this fact).

Additional comments

The revised version of the manuscript presents completely rewritten manuscript which is much more focused and clearer. All the comments from first revision were taking into account.
There are some minor issues that needs to be corrected or additionally explained:

l. 40: term “salmonids and trout species”. Please, explain the difference (trout species are salmonids).
l. 51: “marbled coloration associated with a strong phylogenetic signal.” Please, explain what is meant by this. Marbled coloration is present in i.e. Otra trout population, which is phylogenetically not closely related to S. marmoratus. Some Adriatic and Mediterranean populations of brown trout are characterised by MA mtDNA haplotype. Your statement therefore needs additional explanations.
l. 76: term “Atlantic brown trout”. Please, explain what is meant by this. Maybe brown trout of AT mtDNA phylogenetic lineage?
l. 141-145: are all these trout taxa used for stocking fertile? This fact once again calls into question the presence of marble trout in Toce river and suggests a more likely presence of hybrids among all the taxa stocked to the river. Except if there is some kind of ecological barrier to hybridization.
l. 224: “marbled elements”. Since these “elements” are the criteria for selection of samples, please, describe in details, what is meant by this term? How many people have selected the fish? Or was this selection made by only one person? What was the criteria to define a trout as a hybrid and not include it in the study?
Later you refer to Figure S1 where some marble trout are presented. The fish shown are all small (and maybe also the photos are stretched out?), not sexually matured and therefore, not displaying an adult pigmentation. It is therefore hard to say that these are “pure” marble trout with all “marbled elements” present. Please, add photos of sexually matured fish (TL>55cm) which display adult pigmentation. It would also benefit to add some photos of specimens from Italy that were according to genetic analyses determined as pure to see the real "marbled elements".
l. 382: only here you mention collection of marble trout and hybrids. This fact should be stressed before, so that the reader is aware of it. Please, see also comment above about the Fig. S1.
l. 393: Fig. S4. Phenotypes of presented fish do not completely represent the marble trout phenotypes. To my opinion, these fish are most probably hybrids.
l. 404: are there any differences between the sections of Toce river considering the growth trajectory?
Discussion: you mention some limitations of this and other studies that phenotypically discriminated between ‘pure’ and hybrid individuals. But as I understand, you have the genetic data for your population (Polgar et al., submitted), therefore, you could easily bypass this limitation and prepare a better and more credible study.

---

## Round 0.3 · accepted · Accept

Thanks for addressing all of the reviewers' comments. Congratulations!